# Real-world ethics in palliative care: protocol for a systematic review of the ethical challenges reported by specialist palliative care practitioners in their clinical practice

Guy Schofield,[1] Emer Brangan,[2] Mariana Dittborn,[3] Richard Huxtable,[1] Lucy Selman[4]

¹Centre for Ethics in Medicine, University of Bristol, Bristol, UK
²The National Institute for Health Research Collaboration for Leadership in Applied Health Research and Care West (NIHR CLAHRC West), University Hospitals Bristol NHS Foundation Trust, Bristol, UK
³Florence Nightingale Faculty of Nursing, Midwifery & Palliative Care, King's College London, London, UK
⁴School of Social and Community Medicine, University of Bristol, Bristol, UK

**Correspondence to**
Dr Guy Schofield;
guy.schofield@bristol.ac.uk

## ABSTRACT

**Introduction** Ethical issues arise daily in the delivery of palliative care. Despite much (largely theoretical) literature, evidence from specialist palliative care practitioners (SPCPs) about real-world ethical challenges has not previously been synthesised. This evidence is crucial to inform education and training and adequately support staff. The aim of this systematic review is to synthesise the evidence regarding the ethical challenges which SPCPs encounter during clinical practice.

**Methods and analysis** We will conduct a systematic review with narrative synthesis of empirical studies that use inductive methods to describe the ethical challenges reported by SPCPs. We will search multiple databases (MEDLINE, Philosopher's Index, EMBASE, PsycINFO, LILACS, WHOLIS, Web of Science and CINAHL) without time, language or geographical restrictions. Keywords will be developed from scoping searches, consultation with information specialists and reference to key systematic reviews in palliative care and bioethics. Reference lists of included studies will be hand-searched. 10% of retrieved titles and abstracts will be independently dual screened, as will all full text papers. Quality will be dual assessed using the Mixed-Methods Appraisal Tool (2018). Narrative synthesis following Popay *et al* (2006) will be used to synthesise findings. The strength of resulting recommendations will be assessed using the Grading of Recommendations Assessment, Development and Evaluation approach for qualitative evidence (GRADE-CERQual).

**Ethics and dissemination** As this review will include only published data, no specific ethical approval is required. We anticipate that the systematic review will be of interest to palliative care practitioners of all backgrounds and educators in palliative care and medical ethics. Findings will be presented at conferences and published open access in a peer-reviewed journal.

**Trial registration number** CRD42018105365.

### Strengths and limitations of this study

► The systematic review search strategy uses a broad range of electronic databases, including those which index philosophical as well as clinical research and international publications.
► This global review benefits from no language, time or location restrictions in the search strategy.
► The use of peer-reviewed filters for qualitative and survey-based methodologies may lead to loss of some relevant studies.
► The exclusion of studies investigating single ethical issues, such as palliative sedation, risks reducing the depth of detail that will be incorporated into the final synthesis.
► However, the benefit of including only inductive studies reporting specialist palliative care practitioner (SPCP) real-world experiences is that the resultant synthesis will represent only those topics that are directly reported by SPCPs, and therefore better reflect the real-world context of their practice.

aims to maximise quality of life.[1] The focus of care includes both the patient and those close and important to them, such as their family.

Despite the increasing global provision of palliative care services, the need for palliative care is growing and unmet.[2 3] In 2011, 74% of countries worldwide had either no or only isolated palliative care services.[4] The 2017 Lancet Commission Report estimated that globally 61.1 million people required specialist palliative care input in 2015.[5] The majority of these people live in low-income and middle-income countries, where provision of specialist palliative care is highly variable; globally, it is estimated that only 14% of those who might benefit from palliative care receive it.[2] In the UK, modelling predicts that by 2040 there will be both an increase in the absolute numbers of deaths and, due to multimorbidity and medical complexity,

## INTRODUCTION

Palliative care is a holistic approach to the care of patients with life-limiting illness that

an increase in the percentage of those dying that require specialist palliative care.[6] As the current worldwide epidemic of non-communicable diseases grows, this trend is likely to be replicated.[7 8]

In the theoretical literature, palliative care is frequently connected with moral problems across a wide variety of aspects of clinical care.[9] These include, for example, withdrawing and withholding of interventions,[10] dignity and quality of care,[11] respect for autonomy[12] and palliative sedation.[13 14] However, there is evidence from other areas of healthcare practice that the ethical dilemmas discussed in the literature do not accurately reflect the range of the dilemmas that healthcare workers report experiencing in real-world practice.[15–17] While this phenomenon of a mismatch between lived experience and the academic literature has not previously been systematically examined within palliative care, there is some evidence suggesting it does apply.[18–20] Hermsen and ten Have,[18] for example, compared the ethical challenges reported by specialist palliative care practitioners (SPCPs) from the Netherlands with those found in the palliative care literature. They found 14 reported ethical challenges with no accompanying literature, and two topics with significant literature (organ donation and engagement with ethical committees), but which were not reported in practice.[18]

We aim to address this knowledge gap by systematically reviewing and synthesising the published evidence regarding the ethical challenges reported by SPCPs, in order to generate an understanding of these real-world challenges. This is crucial to the specialty going forward: the need for training in the ethical aspects of palliative care is recognised as a priority,[21] and a thorough understanding of the ethical context practitioners work within is needed if educators are to generate evidence-based curricula that reflect real world contexts. Education benefits from a robust grounding in the real-world experiences of learners: the relevance of educational material is a key factor in adult learner motivation,[22] and processing new material in relation to prior experiences contributes to learning efficiency.[23] Similarly, in the field of bioethics, there has recently been an 'empirical turn', central to which is the idea that understanding the real-world context of moral problems is a key part of their analysis.[24] Fundamental to high-quality empirical bioethics is an accurate understanding of context, taking robust empirical evidence as a starting point.[25]

## Aim

We aim to systematically review the literature to answer the following research question: what do SPCPs report as ethical challenges that they experience during clinical practice?

## METHODS AND ANALYSIS
### Eligibility criteria

This review aims to identify studies that describe the ethical challenges reported by SPCPs in their day-to-day clinical practice. The inclusion and exclusion criteria are summarised in table 1. Strech *et al* describe an adaptation of the population, intervention, comparison and outcome (PICO) system for systematic reviews that are examining empirical bioethical topics;[26] we use their Methodology, Issue, Participants (MIP) system.

The review will include peer-reviewed studies in which SPCPs report the ethical challenges they face in their real-world clinical practice or secondary analyses of such data. Studies must derive their data using inductive methods; deductive research, in which researchers prespecify a priori the ethical challenges they focus on, will be excluded. Following Creswell and Plano Clark,[27] deductive research 'works from the "top down", from a theory to hypotheses to data to add to or contradict the theory', while inductive research is 'bottom-up, using the participants' views to build broader themes and generate a theory interconnecting the themes' (p. 23).[27] We consider inductive data as deriving from data collection efforts that are independent from any attempt to validate a particular theory or hypothesis. While much inductive data are qualitative or mixed methods by design, it can also include quantitative studies (eg, surveys using questionnaire items originally derived inductively using qualitative methods rather than specified a priori).

We will include only inductive studies as we aim to generate a landscape of challenges experienced in the real-world context. Scoping searches identified multiple studies investigating preselected ethical challenges within the practice of palliative care. Studies of this type will be excluded as they reflect choices of the study authors rather than the real-world experience of practitioners; including them in the synthesis would risk introducing data that do not reflect SPCPs experiences.

Similarly, studies that explore single ethical challenges in specialist palliative care practice, for example, palliative sedation or advance care planning, will be excluded. These studies proceed from an a priori assumption that their topic of interest is present in the real-world experience of SPCP's. Excluding them will therefore minimise the risk of introducing ethical challenge data that is not present in the real-world experience of SPCPs.

Non-peer reviewed papers, studies not reporting inductive empirical data, book chapters, editorials and theses, case reports, opinion pieces and reviews will be excluded.

There will be no language or timeframe restrictions.

## Search strategy
### Electronic searches

The following databases, identified in conjunction with subject information specialists and indexing journals containing key papers known to the research team, will be searched: MEDLINE (Ovid interface, 1946 onwards), Philosopher's Index (OVID interface 1940 onwards), EMBASE (OVID interface, 1980 onwards), PsycINFO (OVID interface 1806 onwards), LILACS (http://lilacs.bvsalud.org/en/ 1982 onwards), Web of Science

| Table 1 | Inclusion and exclusion criteria | |
|---|---|---|
| | **Inclusion criteria** | **Exclusion criteria** |
| Types of participants | Study participants are SPCPs in a patient care role. We define SPCPs as people working in, or for, a healthcare setting whose main focus is on delivering palliative care (as opposed to clinical contexts where palliative care forms part, but not the main focus, of the care provided). This may include (but is not limited to) nurses, doctors, occupational therapist, physiotherapists, dieticians, speech and language therapists, psychologists, other allied health professionals and chaplains.<br>Studies with a mixed population where SPCP participants' data are separately presented and can be extracted will be included. | Participants who undertake palliative care tasks as part of their role (eg, oncologists), but who do not specialise in providing palliative care and do not have palliative care as the main focus of their role. |
| Context | All geographical settings and all clinical settings where SPC is delivered will be included. | Studies conducted in settings in which SPC is not being delivered. |
| Issues | The range of ethical challenges that are reported as experienced by SPCPs during clinical delivery of palliative care.<br>The definition of 'ethical challenges' will be intentionally kept broad to capture the maximum number of examples. It includes but is not limited to terms such as ethical issues, moral challenges, moral dilemmas, values, good/bad, right/wrong. Ethical challenges can be labelled as such either by authors or participants. | Studies that use survey tools with preselected ethical dilemmas that have not been inductively derived based on evidence from SPCPs, and studies that investigate a single aspect of palliative care only will be excluded.<br>These study designs are excluded as they proceed from an a priori assumption that their selected issues are relevant. They, therefore, do not contribute to an inductive exploration of the breadth and type of ethical challenges facing practitioners. |
| Methodologies | Empirical studies examining, using inductive methods, the ethical challenges reported by SPCPs in their clinical practice. These may include qualitative studies, mixed methods studies (eg, surveys with free-text responses) or quantitative studies using questionnaires derived inductively through consultation with SPCPs. | Studies not reporting inductively derived empirical data. These may include studies using questionnaires which include ethical challenges selected a priori or single-issue studies focused on an ethical challenge selected a priori by the researchers. |
| Timeframe | Any time frame up until the search date will be included, contingent on the inception dates of the databases included in the search. | |
| Type of publications | Peer-reviewed journal publications of empirical research. Papers in any language will be included, with findings translated into English where necessary. | Where no full text is available through the university subscription, study authors will be contacted for full text. If there is no response within 2 weeks, the study will be excluded.<br>The following will also be excluded:<br>▶ Conference abstracts; however, authors will be contacted for further data/publications.<br>▶ Editorials, letters or comment/opinion pieces.<br>▶ Review articles. Reviews will be used for identification of primary research only.<br>▶ Book sections. |

SPCP, specialist palliative care provider.

(Clarivate interface, 1900 onwards) and CINAHL (EBSCO interface, 1937 onwards). There will be no language, geographical or time limits. Non-English-language records will be screened by a native speaker of the relevant language. If a non-English-language paper is included in the review, it will be translated into English prior to integration in the analysis.

Initial search terms were developed with reference to the key words of major systematic reviews in palliative care and bioethics. Scoping searches suggested that the initial search terms would result in over 20 000 records returned and that relevant studies would be qualitative (eg, using interviews or focus groups) or use survey-based methodologies. To increase the specificity of the search, we will therefore apply peer-reviewed methodological filters for these study designs, identified via the InterTASC Information Specialists' Sub-Group Search Filter Resource.[28] The MEDLINE search strategy (see online supplementary file 1) will be checked and modified for the other databases as appropriate.

### Searching other resources

Reference lists of included papers will be hand-searched. Corresponding authors of papers meeting the inclusion criteria will be contacted to ascertain if there are other published papers they recommend for review. Authors of conference abstracts will be contacted for peer-reviewed data or follow-up publications if available; both will be included if provided and eligible. Papers that cite the included studies will be screened for inclusion.

A grey literature search will not be conducted. Cook *et al* demonstrated that an extensive grey literature search did not benefit the review content of a palliative care systematic review despite the significant resources required to undertake it.[29]

### Selection process

All titles and/or abstracts of retrieved records will be screened to identify papers that potentially meet the inclusion criteria. The first researcher (GS) will screen the full search results. A second researcher (MD) will independently screen a random sample of 10%. Differences in screening between GS and the second reviewer will be discussed with the research team to clarify and refine inclusion/exclusion criteria. Contested papers will be discussed and any that remain unresolved will be examined by third reviewer (LS).

The full text of potentially eligible records will be retrieved and independently assessed for eligibility by two review team members (GS, MD). Any disagreement between them over the eligibility of particular papers will be resolved through discussion with a third reviewer (LS).

### Data extraction and management

Search results will be exported and collated in Endnote X8. Records will be deduplicated and numerical results will be recorded and presented in a flowchart that follows the PRISMA design.[30]

Data extraction will be undertaken independently by two reviewers, using a prepiloted data extraction form. Disagreements will be resolved through consultation with a third reviewer if necessary. Data items to be extracted from included studies will include: (1) citation details including title, publication year and journal; (2) study setting, methods, participant characteristics, sample size; (3) specified definition/conceptualisation of ethical challenges; (4) key findings, themes and subthemes; (5) sources of potential bias including funders and evidence of reflexivity. In the event of relevant missing data, corresponding authors will be contacted.

### Data synthesis

We will undertake a systematic narrative synthesis, following the iterative framework proposed by Popay *et al*,[31] adapted for a review which does not focus on an intervention: (1) developing a preliminary synthesis of study findings, (2) exploring relationships in the data, (3) assessing the robustness of the synthesis product and (4) developing a theoretical model of ethical challenges in the real-world practice of SPCPs. Stage 1 will include integrating the themes and content of qualitative studies; this will be guided by the 'thematic synthesis' approach developed by Thomas and Harden.[32] The narrative synthesis will explore findings within and across included studies, taking into account study quality (see below); identify patterns in the data and synthesise the described ethical challenges in an overarching framework or model.

### Risk of bias (quality) assessment

Scoping searches suggest that multiple study designs may be returned. So that the quality of diverse study designs can be compared, we will use the Mixed-Methods Assessment Tool (MMAT) (2018 Version)[33] which allows for comparison of quality between studies using differing methodologies. We will not use low MMAT scores to exclude studies, but we will reflect on study quality and the effect of lower scoring studies on the resulting synthesis. Two reviewers (GS, MD) will score each of the included studies independently. Any disagreements will be resolved by consulting a third independent reviewer.

While this review is not designed to produce recommendations for clinical practice, it is nevertheless important that we reflect on our confidence in the evidence synthesis. As the focus of the review is on inductively derived empirical data, we will use the GRADE-CERQual framework to do so.[34] CERQual provides a systematic and transparent framework for assessing confidence in individual review findings, based on consideration of four components: (1) methodological limitations, (2) coherence, (3) adequacy of data and (4) relevance. Assessments of the four components collectively contribute to an overall assessment of whether findings from a qualitative evidence synthesis provide a reasonable representation of the phenomenon of interest.

### Ethics and dissemination

As this review will include only published data, no specific ethical approval is required.

This systematic review will synthesise empirical evidence on the ethical challenges reported by SPCPs. The research team anticipate that it will be of interest to palliative care practitioners of all backgrounds, and educators involved in palliative care or postgraduate ethics training. Findings will be presented at relevant conferences and published in a peer-reviewed journal in open access format.

### Patient and public involvement

Patients and the public were not involved in designing the protocol of this systematic review.

## DISCUSSION

Ethical challenges are a significant part of the day-to-day experience of working as a SPCP. This systematic review will, to our knowledge, be the first to synthesise studies that examine practitioner-reported challenges. We hope that better understanding the ethical challenges experienced

by healthcare practitioners working in palliative care in their day-to-day practice will help to inform:

1. Palliative Care Education. This synthesis of the evidence will help identify ethics training needs and inform educational training curricula for all those involved in palliative care provision.
2. Clinical Ethics Education. This review will further develop the evidence base that supports design of more general ethics curricula (eg, for philosophers, lawyers or social scientists working in or learning bioethics), including revision of the topics included in these curricula and critical examination of the assumptions behind these choices.
3. Research. This work will establish the state of the science in this field and provide a sound basis on which to identify palliative care bioethics research priorities.

The protocol design decisions we have made are associated with potential limitations. First, the search strategy uses methodological filters. While this accords with Strech *et al*'s recommendation that empirical bioethics reviews limit the number of methodologies that are included,[26] this approach may filter out studies that contain relevant data. Pilot searches were evaluated for study loss using studies known about prior to the review; all were returned by the search strategy. Additional search strategies (handsearching reference lists and contacting authors of included studies) will also be employed. However, it is possible that a relevant study might not be identified due to misclassification in the registry or use of another relevant methodology in a novel way.

Second, our criteria exclude studies that are not inductive in nature, to ensure we capture the 'real-world' challenges of clinical practice and mitigate potential bias towards using Western ethical principles as a means of structuring and collecting data on ethical challenges, for example, in questionnaires. We use authors' descriptions of study design to determine whether the study reported used an inductive or deductive approach. However, even in purely qualitative research, data collection can be structured to varying degrees; this is difficult to determine without access to the raw data used in analyses. Notwithstanding this limitation, our inclusion and exclusion criteria are designed to exclude those studies which specifically selected a priori which topics were of interest and hence did not allow flexibility in terms of the ethical challenges raised by participants.

Third, we also exclude studies which focus on the ethical challenges of a particular aspect of palliative care, for example, the ethical challenges within palliative sedation or advance care planning. Studies that focus on particular aspects of practice are likely to generate granular data about particular challenges. This level of data would allow for better understanding of the complex nature of these topics. However, in their comparison between observed ethical challenges and the content of the palliative care ethics literature, Hermsen and ten Have demonstrate that the topics selected by authors for investigation in this manner may not represent the challenges that are faced

in real-world practice.[18] The inclusion of single issue studies would increase the risk of this occurring in this review. To meet our aim of developing a model of ethical challenges based on real-world practice, we will therefore exclude these studies.

Finally, quality assessment of qualitative research is a contested area, with multiple tools available and often poor correlation between methods.[35] The MMAT contains fewer criteria to assess study quality than methodology specific tools, for example, the CASP Qualitative Check List.[36] This may lead to an incorrect overassessment or underassessment of a study's inherent bias. However, as we will not exclude studies based on their MMAT scores, we believe the ability to directly compare studies of differing methodologies has significant benefits in terms of utility to this review.

## Reporting

This study protocol has been designed with reference to Preferred Reporting Items for Systematic Reviews and Meta-Analyses Protocols (PRISMA-P)[30] (see online supplementary file 2 for checklist). The review will be reported in line with the Preferred Reporting Items for Systematic Reviews and Meta-Analyses (PRISMA) statement.[37]

**Acknowledgements** The authors would like to thank Alison Richards and Sarah Herring for their assistance in developing the search strategies for this review, and Huey Yen Chia for their help translating Traditional Chinese language records for screening.

**Contributors** GS, LS, EB and RH conceived of the review and developed the protocol. GS, EB, MD and RH developed the search strategy. GS and LS wrote the manuscript draft. All authors revised and edited the draft manuscript and approved the final version.

**Funding** This work was supported by Wellcome Trust Research Award for Health Professionals (208129/Z/17/Z) for GS. The sponsor is the University of Bristol.

**Competing interests** None declared.

**Patient consent for publication** Not required.

**Provenance and peer review** Not commissioned; externally peer reviewed.

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
