## [Reviewer comments · BMJ Open]

ARTICLE DETAILS

TITLE (PROVISIONAL)	Real-world Ethics in Palliative Care: Protocol for a Systematic Review of the Ethical Challenges Reported by Specialist Palliative Care Practitioners in their Clinical Practice
AUTHORS	Schofield, Guy; Brangan, Emer; Dittborn, Mariana; Huxtable, Richard; Selman, Lucy

VERSION 1 - REVIEW

REVIEWER	Ghislaine van Thiel University Medical Center Utrecht The Netherlands.
REVIEW RETURNED	05-Jan-2019

GENERAL COMMENTS	I have reviewed the protocol with great interest and appreciate the effort to design a review protocol for ethical issues in a field that is in itself challenging to research. In general the protocol is well described. In my opinion however, several aspects are not well enough specified in order to make the review successful. I have the following points and questions: 1. The rationale for the study on p4 clearly describes the relevance of palliative care and the importance of empirical evidence to generate knowledge most relevant to the real world context. However, the section on the ethical issues could in my opinion be more convincing (p4 line 28-40): why are ethical challenges a particular concern in palliative care? What exactly is the knowledge gap the authors want to address?2. The authors describe they only want to include inductive studies. There is no clear definition given. If I understand correctly, the idea is that in these studies the most relevant experience of ethical challenges can be found. The current description of in- and exclusion criteria may not be specific enough to guide the study selection. Surveys with free text options as well as qualitative research using for example semi-structured interviews may be included. However, those methods may comprise both spontaneous self-reporting of ethical challenges as well as questions from the researcher about specific issues. Especially quantitative surveys generally offer little room for self reporting. The fact that a free text option is required for inclusion is in my opinion not sufficient to claim only inductive research is included. In addition, even more open, qualitative methods do not necessarily imply the level of self reporting the authors seem to aim for (for example because the interview guide may be quite directive). The authors are very clear that they do not want reports on pre-selected issues. It may be very difficult to
---

identify in which studies such pre-selection was performed. How will the authors assess the level of 'spontaneous' self reporting and guidance towards specific issues by the researchers?

3. On P11 the authors explain their choice to exclude single-issues studies. Their argument is not convincing to me: the fact that single issue reports do not represent a broad range of challenges is of course true, but at the same time it is no argument for excluding single issue studies from this review. The single issues mentioned (i.e. palliative sedation) may cover a substantial part of palliative care and the studies on these issues may reveal self-reported ethical challenges that should not be overlooked.

4. It is insufficiently clear to me throughout the manuscript what type of data the review is aimed at. I think this should be explained better. For example: is the review about experiences of SCPCs? If so, this should be specified in the data extraction. Furthermore, when does an experience/challenge qualify as a day-to-day challenge?

5. Regarding the definition of the study population: "people working in or for a health care setting whose main focus is on delivering palliative care (as opposed to clinical contexts where palliative care forms part, but not the main focus, of the care provided)" (Table 1). Strictly speaking this also involves people who are not directly involved in the delivery of palliative care i.e. hospital management. I assume this is not what the authors really want?

6. Since palliative care is often combined with other care, it is wise to exclude professionals who provide palliative care as part of their role and work for example as oncologist in a hospital? Is it possible that important ethical challenges (i.e. in discussing palliative care with patients) will be overlooked?

Minor questions and points:

7. On P5 : the authors state that "(...), there is evidence from other areas of healthcare practice that the ethical dilemmas that are written about in the literature do not reflect the range of the dilemmas that healthcare workers report experiencing on a day-to-day basis.[21–23]" This is a protocol for a review and this statement raises the question whether the experience the authors are looking for is at all present in the literature. How should we read this?

8. In the Aim on p5. the day-to-day aspect is omitted. Why did the authors choose this?

9. P 3 Strength: "However, the benefit of including only inductive studies is that the resultant synthesis will represent only those topics that are directly reported by SPCPs, reflecting the realworld context." I wonder if this is correct (this may be a language issue): also in non-inductive research (such as structured interviews) topics may be directly reported?

10. In the Methods no language restrictions are anticipated. I was wondering if that is feasible? On P6 it seems to be more nuanced: Papers in any language will be included, with findings translated into English. But if only findings are translated, how will the authors assess the inductive nature of a study if they do not master the paper's language?

11. P6: Study type and data should be reported consistently throughout the manuscript. Currently, for example, on p6 it says the data should not only come from inductive studies, but should also be primary data. It is not clear to me what this adds?

	12. For outcome listing (Item 13 of PRISMA-P) the authors refer to Table 1. As far as I can see this Table does not list outcomes? At least, it does not specify which outcomes are considered to be reflective of self-reported day-to-day ethical challenges.
--	---

REVIEWER	Prof Nancy Preston International Observatory on End of Life Care Faculty of Health and Medicine Lancaster University Lancaster LA1 4YG UK
REVIEW RETURNED	29-Jan-2019

GENERAL COMMENTS	This is a clear protocol. Some areas to clarify would be: 1. Page 4 lines 29-32 in addition to the ethical dilemmas mentioned wouldn't hasten death (especially in countries where legalised or permissible such as Benelux countries, US states and Switzerland etc) and double effect also be major ethical dilemmas? Page 5 lines 3-6 I am unclear what you mean 'not referenced in these fields' means. Double effect and hastened deaths are in most bioethics textbooks for example or are you excluding these topics? Page 6 lines 17-29 I would make clear that not all quantitative studies (maybe remove quantitative) and just leave that free text parts of qualitative studies. This was only clear after reading most of the paper) Page 10 lines 5-8 in regards Popay's Narrative Synthesis. Rather than referring to these as stages they are a framework and the framework is iterative - not stages and sequential. In addition, the framework includes developing theory - could you justify why you are omitting this part of the framework? The main issue in the this review is that the authors choose to just include inductive papers which I interpret to mean papers that are based on broad interview questions that almost happen to include ethical or moral discussions. They recognise the limitations of this as they exclude papers with a single focus ie palliative sedation and this could be a real weakness as in these interviews other topics would quite naturally be discussed such as euthanasia (especially in the Benelux countries) and double effect of treatments such as opioids.
---

VERSION 1 – AUTHOR RESPONSE

Reviewer 1	
I have reviewed the protocol with great interest and appreciate the effort to design a review protocol for ethical issues in a field that is in itself challenging to research. In general the protocol is well described. In my opinion however, several aspects are not well enough specified in order to make the	Thank you.

review successful. I have the following points and questions:	
1. The rationale for the study on p4 clearly describes the relevance of palliative care and the importance of empirical evidence to generate knowledge most relevant to the real world context. However, the section on the ethical issues could in my opinion be more convincing (p4 line 28-40): why are ethical challenges a particular concern in palliative care? What exactly is the knowledge gap the authors want to address?	Thank you for this comment. We have updated the text of the introduction to clarify the knowledge gap and how this review seeks to address it. We would contend that all fields of healthcare contain significant and varied ethical challenges. Palliative care is not 'special' in its level of challenges or their complexity. We hope this methodology will be applicable to other fields and the results will demonstrate the value of this approach.
2. The authors describe they only want to include inductive studies. There is no clear definition given. If I understand correctly, the idea is that in these studies the most relevant experience of ethical challenges can be found. The current description of in- and exclusion criteria may not be specific enough to guide the study selection. Surveys with free text options as well as qualitative research using for example semi-structured interviews may be included. However, those methods may comprise both spontaneous self-reporting of ethical challenges as well as questions from the researcher about specific issues. Especially quantitative surveys generally offer little room for self reporting. The fact that a free text option is required for inclusion is in my opinion not sufficient to claim only inductive research is included. In addition, even more open, qualitative methods do not necessarily imply the level of self reporting the authors seem to aim for (for example because the interview guide may be quite directive). The authors are very clear that they do not want reports on pre-selected issues. It may be very difficult to identify in which studies such pre-selection was performed. How will the authors assess the level of 'spontaneous' self reporting and guidance towards specific issues by the researchers?	We define inductive and deductive research following Creswell and Plano Clark 2007[1]: deductive research "works from the 'top down', from a theory to hypotheses to data to add to or contradict the theory" while inductive research is "bottom-up, using the participants' views to build broader themes and generate a theory interconnecting the themes" (p. 23). We now reference this definition on Page 6. Crucially, inductive research is concerned with moving from the specific (e.g. experiences, observations) to the general (e.g. principles or hypotheses). This type of 'specific' data is what we aim to capture in this review. Our inclusion of inductive research aims to mitigate the bias that is present in ethical literature, i.e. a bias towards using specific Western ethical principles as a means of structuring and collecting data on ethical challenges, e.g. in questionnaires. We therefore include qualitative research which adopts an inductive, exploratory approach, allowing participants to "speak for themselves", as well as mixed-methods approaches with a similar component (e.g. questionnaires which allow free-text comments or stating issues which have been derived from prior qualitative research). We agree with the reviewer that there is of course variation in the extent to which qualitative research is itself 'structured' and that this is difficult to determine without access to the raw data used in analyses – we now mention this in our Limitations section on page 13. However, our inclusion and exclusion criteria

	are designed to exclude those studies which specifically selected a priori which topics are of interest and hence do not allow flexibility in terms of the challenges raised by participants.
3. On P11 the authors explain their choice to exclude single-issues studies. Their argument is not convincing to me: the fact that single issue reports do not represent a broad range of challenges is of course true, but at the same time it is no argument for excluding single issue studies from this review. The single issues mentioned (i.e. palliative sedation) may cover a substantial part of palliative care and the studies on these issues may reveal self-reported ethical challenges that should not be overlooked.	We have now included a more detailed explanation of the rationale for this decision (pages 6-7). We accept it is a balance between the risk of overlooking an important finding in a single-issue study, and introducing challenges that do not form part of real-world experience by SPCPs. However, as our primary concern in the compromise is to avoid the accidental inclusion of ethical challenges that do not form part of practitioners' real-world experience, we have chosen to exclude these studies.
4. It is insufficiently clear to me throughout the manuscript what type of data the review is aimed at. I think this should be explained better. For example: is the review about experiences of SCPCs? If so, this should be specified in the data extraction. Furthermore, when does an experience/challenge qualify as a day-to-day challenge?	Yes, this review is about the experiences of SPCPs – we have now explicitly stated this on page 5 and this is repeated in Table 1. We have changed the phrasing of 'day-to-day' to 'real-world' We hope this better captures the idea that we are examining ethical challenges experienced in real-world practice, rather than academic or abstract ones. Most will overlap of course.
5. Regarding the definition of the study population: "people working in or for a health care setting whose main focus is on delivering palliative care (as opposed to clinical contexts where palliative care forms part, but not the main focus, of the care provided)" (Table 1). Strictly speaking this also involves people who are not directly involved in the delivery of palliative care i.e. hospital management. I assume this is not what the authors really want?	We have adjusted this section to make it clearer that the review focuses on those staff members with patient contact as part of their role.
6. Since palliative care is often combined with other care, it is wise to exclude professionals who provide palliative care as part of their role and work for example as oncologist in a hospital? Is it possible that important ethical challenges (i.e. in discussing palliative care with patients) will be overlooked?	We are aiming to synthesise the challenges reported by individuals working in specialist palliative care. While it is also important to understand the ethical challenges experienced by other clinicians such as oncologists who could be seen as providing 'generalist' palliative care as part of their role, that is not the focus of this review. By focusing on the speciality of palliative care, we hope to elucidate issues related to its underlying philosophy, principles and guidelines which are not shared across all providers of end of life care. Including non-specialist palliative care providers in the review

	would therefore risk diluting the relevance of the findings to the field of specialist palliative care.
Reviewer 1 Minor	
7. On P5 : the authors state that “(...), there is evidence from other areas of healthcare practice that the ethical dilemmas that are written about in the literature do not reflect the range of the dilemmas that healthcare workers report experiencing on a day-to-day basis.[21–23]” This is a protocol for a review and this statement raises the question whether the experience the authors are looking for is at all present in the literature. How should we read this?	Two studies that meet the inclusion criteria were known prior to designing the protocol.[2,3] Furthermore, scoping searches revealed other records that meet the inclusion criteria. We tracked these records during the identification and translation of the methodological filters and they were successfully retrieved using the search strategy employing filters. We were therefore confident at the outset that there were published papers regarding SPCP's experiences of ethical challenges that would benefit from synthesis.
8. In the Aim on p5. the day-to-day aspect is omitted. Why did the authors choose this?	See response to point 4 above. We have changed the references from 'day-to-day' to 'real-world'.
9. P 3 Strength: “However, the benefit of including only inductive studies is that the resultant synthesis will represent only those topics that are directly reported by SPCPs, reflecting the realworld context.” I wonder if this is correct (this may be a language issue): also in non-inductive research (such as structured interviews) topics may be directly reported?	Please see our response to point 2 above.
10. In the Methods no language restrictions are anticipated. I was wondering if that is feasible? On P6 it seems to be more nuanced: Papers in any language will be included, with findings translated into English. But if only findings are translated, how will the authors assess the inductive nature of a study if they do not master the paper's language?	We have resources to ensure that non-English-language records are handled in the same way as English records. Findings will be translated into English prior to analysis. We have now clarified this on page 7.
11. P6: Study type and data should be reported consistently throughout the manuscript. Currently, for example, on p6 it says the data should not only come from inductive studies, but should also be primary data. It is not clear to me what this adds?	Thank you for this comment. We have removed the reference to primary data.
12. For outcome listing (Item 13 of PRISMA-P) the authors refer to Table 1. As far as I can see this Table does not list outcomes? At least, it does not specify which outcomes are	Thank you for pointing out this anomaly, which highlights the difficulty of fitting a systematic review of this nature (i.e. on ethical challenges) into the PRISMA-P system. Motivated by this comment, we re-examined our approach and

considered to be reflective of self-reported day-to-day ethical challenges.	now use the adaptation of the PICO system developed for empirical bioethics reviews: Strech et al's Methodology, Issue, Participants system [4]. This is detailed on page 5 and in table 1.
Reviewer 2	
This is a clear protocol. Some areas to clarify would be:	Thank you
1. Page 4 lines 29-32 in addition to the ethical dilemmas mentioned wouldn't hasten death (especially in countries where legalised or permissible such as Benelux countries, US states and Switzerland etc) and double effect also be major ethical dilemmas?	In describing dilemmas, we chose as examples those that were applicable to palliative care globally. Most clinicians see euthanasia and assisted suicide as outside of the scope of specialist palliative care practice. In some countries euthanasia does lie alongside palliative care, in that other clinicians will undertake these practices but often need the input of palliative care. However, we don't believe this necessitates the inclusion of euthanasia as one of the palliative care challenges listed here.
Page 5 lines 3-6 I am unclear what you mean 'not referenced in these fields' means. Double effect and hastened deaths are in most bioethics textbooks for example or are you excluding these topics?	We have updated the relevant sentence to clarify our meaning. This sentence related to the lack of use of empirically-derived ethical evidence in the clinical ethical discussions and in the design of educational materials.
Page 6 lines 17-29 I would make clear that not all quantitative studies (maybe remove quantitative) and just leave that free text parts of qualitative studies. This was only clear after reading most of the paper)	We are sorry but we are unclear as to what this refers to. We refer to our response to point 2 above regarding our inclusion of inductive studies (using qualitative or mixed methods methodology, e.g. free-text responses within structured questionnaires).
Page 10 lines 5-8 in regards Popay's Narrative Synthesis. Rather than referring to these as stages they are a framework and the framework is iterative - not stages and sequential. In addition, the framework includes developing theory - could you justify why you are omitting this part of the framework?	Thank you – we have now adjusted the text to emphasise the iterative nature of our approach and our aim of developing a theory or model of real-world ethical challenges (page 11).
The main issue in the this review is that the authors choose to just include inductive papers which I interpret to mean papers that are based on broad interview questions that almost happen to include ethical or moral discussions. They recognise the limitations of this as they exclude papers with a single focus ie palliative sedation and this could be a real weakness as in these interviews other topics would quite	Please see our response to point 2 above.

naturally be discussed such as euthanasia (especially in the Benelux countries) and double effect of treatments such as opioids.	
--	--

VERSION 2 – REVIEW

REVIEWER	Ghislaine van Thiel University Medical Center Utrecht The Netherlands
REVIEW RETURNED	03-Mar-2019

GENERAL COMMENTS	The authors convincingly address the points of my review. I am looking forward to the results of the systematic effort, which I believe will enrich the ethical debate on relevant issues in (palliative) care.
---

REVIEWER	Nancy Preston Lancaster University
REVIEW RETURNED	11-Mar-2019

GENERAL COMMENTS	Whilst the authors have responded to the queries raised I feel the comment about inductive research is not fully addressed and is confusing. I think including 'qualitative research' rather than 'inductive research' would be clearer and far easier to apply and then be replicable.
---

VERSION 2 – AUTHOR RESPONSE

We thank the reviewers for their ongoing consideration of the revised version of the above paper. We are pleased to have adequately addressed the concerns of reviewer 1. We would like to respond to the ongoing reservations of reviewer 2, namely:

“Whilst the authors have responded to the queries raised I feel the comment about inductive research is not fully addressed and is confusing. I think including 'qualitative research' rather than 'inductive research' would be clearer and far easier to apply and then be replicable.”

We have chosen to use the term 'inductive' research, not 'qualitative', as 'inductive' captures all the study designs we are interested in synthesising for the purposes of this review. The scoping searches we conducted in developing the protocol identified mixed-methods and quantitative studies that meet the inclusion criteria and contain valuable data for the review synthesis. For example, the following studies are not qualitative but include inductively generated data on the ethical challenges experienced by specialist palliative care practitioners:

Chiu T-Y. Ethical dilemmas in palliative care: a study in Taiwan. *J Med Ethics* 2000;26:353–7.
doi:10.1136/jme.26.5.353

Hernández-Marrero P, Pereira SM, Carvalho AS. Ethical Decisions in Palliative Care: Interprofessional Relations as a Burnout Protective Factor? Results From a Mixed-Methods Multicenter Study in Portugal. *Am J Hosp Palliat Med* 2016;33:723–32

These studies are not qualitative by design, but contain relevant data for the mapping of real-world ethical challenges. Adjusting our inclusion criteria to ‘qualitative only’ methodologies would exclude these studies and would lead to the loss of valuable data in the synthesis. We therefore continue to believe the current inclusion criteria are appropriate and meet the right balance between reproducibility and utility in relation to the aims of this review. To further clarify our methods, we have redrafted the relevant paragraph in the manuscript (page 6). In particular, we now state:

“While much inductive data is qualitative or mixed methods by design, it can also include quantitative studies (e.g. surveys using questionnaire items originally derived inductively using qualitative methods rather than specified a priori).”